# The Effects of a Microorganisms-Based Commercial Product on the Morphological, Biochemical and Yield of Tomato Plants under Two Different Water Regimes

**DOI:** 10.3390/microorganisms7120706

**Published:** 2019-12-16

**Authors:** Carmen-Simona Inculet, Gabriela Mihalache, Vincenzo Michele Sellitto, Raluca-Maria Hlihor, Vasile Stoleru

**Affiliations:** 1“Ion Ionescu de la Brad” University of Agricultural Sciences and Veterinary Medicine, 3 M. Sadoveanu, 700440 Iasi, Romania; simona_inculet@yahoo.com (C.-S.I.); rallu_ca@yahoo.com (R.-M.H.); 2Integrated Center of Environmental Science Studies in the North East Region (CERNESIM), The “Alexandru Ioan Cuza” University of Iasi, 700506 Iasi, Romania; 3MsBiotech SPA, Department of Microbiology, Via Zara 23, 00198 Roma, Italy; michele.sellitto@msbiotechspa.com

**Keywords:** organic agriculture, tomato, PGPR, AMF, fungi, irrigation

## Abstract

The practice of organic agriculture represents an essential requirement for conserving natural resources and for providing the food necessary for a growing population, on a sustainable basis. Tomatoes are considered to be one of the most important crops worldwide. In this context, the organic production of tomatoes should be taken into more consideration. The use of microorganisms-based commercial products is an alternative to chemical fertilizers. Anyway, the results of their use are still variable because of various factors. The aim of this study was to test the effect of inoculation with AMF, PGPR and fungi-based products (Rizotech plus^®^) on the morphological (length of the plants), biochemical (lycopen, polyphenols, antioxidant activity), and number of fruits and yields of four tomato cultivars (Siriana F1, HTP F1, Minaret F1, Inima de Bou) in two different water regimes used for irrigation (200 m^3^ or 300 m^3^ of water/hectare) under a protected area. The results showed that the efficiency of Rizotech plus^®^ application is dependent on the cultivar and the amount of water used. Also, it was clearly demonstrated that the microorganism inoculation significantly increased the yield of Minaret F1, Siriana F1 and HTP F1 cultivars as compared to the uninoculated plants, regardless of the water amount used in the experiment. Moreover, it was observed that for the irrigation of all four cultivars, inoculated with Rizotech plus^®^, a lower amount of water (200 m^3^·ha^−1^) can be used to get the same length of plants, number of fruits and yield as in the case of a higher amount of water (300 m^3^·ha^−1^). In the case of lycopene, polyphenols and antioxidant activity, the results varied with the cultivar and the water amount used. This study gives new information about the functionality and performance of the microorganisms from Rizotech plus^®^ product when applied to different tomato cultivars grown in a tunnel, in the condition of two different water regimes, contributing to a better characterization of it and maybe to a more efficient use in agriculture to achieve optimum results.

## 1. Introduction

The decrease of organic crops productivity is directly influenced by the action of biotic and abiotic stress factors. Biotic factors include stresses caused by pathogenic organisms and pests, such as fungi, nematodes, viruses, and insects. Abiotic factors refer to drought, salinity, heavy metals or floods. Yield losses caused by different categories of stress factors can reach up to 50%–82% [1,2]. Worldwide, the organic production of tomato is based on the use of organic nutrients, on antimicrobial compounds obtained from bacteria or plants for combating pests, on the optimal use of water resources for irrigation, and also on the beneficial effects of soil microorganisms [3,4].

Microorganisms such as arbuscular mychorrhizal fungi (AMF), plant growth-promoting rhizobacteria (PGPR) or fungi play a crucial role in stimulating the growth and development of plants via a myriad of direct or indirect mechanisms such as enhanced nutrient acquisition, phytohormones production, induction of systemic resistance in plants, or competing with harmful soil microorganisms. The use of microorganisms in agriculture as biofertilizers represents an eco-friendly alternative to the chemical products that are excessively used in order to obtain high yields, which is also a new approach to the practice of organic farming [5,6,7]. According to FiBL reports, farmers are more and more oriented to organic agriculture. If in 2016, 57.8 million hectares were under organic agricultural management, in 2019, 69.8 million hectares of agricultural land will be managed organically [8]. Tomatoes (*Lycopersicon esculentum* Mill.) are considered very important crops due to their high economic and nutritional values, being a promising crop for organic agriculture. From an economic point of view, the efficiency of tomato culture is ensured by high yields that can exceed 1000 t/ha/year and by technologies that can easily adapt to field, tunnel or greenhouse cultivation, and soil or hydroponic systems. Regarding nutritional values, tomato fruits contain a high variety of nutrients such as carbohydrates (3%); proteins (1.2%); lipids (1%); minerals (calcium, magnesium, phosphorus, potassium, sodium, zinc, manganese); vitamins (vitamins A and C, thiamin, riboflavin, niacin, pantothenic acid and pyridoxine); phenolic acids; flavonoids; and carotenoids, which are also seen as an anticancer agent [9]. Tomato is also a crop known to harbor several microorganisms with root colonizing ability, which, on the one hand, influence directly or indirectly the soil’s health through their beneficial activities and on the other hand, promote the growth and development of plants through their PGPR traits [10]. Therefore, it can be considered suitable for cultivation in organic systems. The beneficial effects of AMF, PGPR or fungi have already been demonstrated in many experiments with species of agronomic interest such as common bean, maize, cucumber or tomato [11]. Also, on the market, different products based on AMF, PGPR or fungi can be found. Over the years, researchers and farmers have complained about their efficiency on the plants because of the variability and inconsistency of field results. Factors such as physical and chemical conditions of the soil, poor ability of the microorganisms to colonize the plant roots, temperature, irrigation, as well as host cultivars have been attributed to the variable responses [12]. Therefore, the emerging commercial products need to be tested under different conditions such as water availability or cultivar variety to have a broader view of its functionality and performance in the organic production of crops such as tomatoes.

In this context, the aim of this study was to test the effects of the inoculation with AMF, PGPR and fungi-based products on the morphological, biochemical, number of fruits, and yield of four tomato varieties under two different water regimes used for irrigation (200 m^3^ or 300 m^3^ of water/hectare).

## 2. Materials and Methods

### 2.1. Plant Material

Four cultivars of *Lycopersicon esculentum* (Siriana F1, Minaret F1, HTP F1, and Inima de Bou) were used in experiments. Three of the cultivars (Siriana F1, HTP F1 and Inima de Bou) have indeterminate growth, while Minaret F1 cultivar has semi-determinate growth. Siriana F1 is an early growing cultivar, known to produce its first fruits after ~100 days. The fruits are red, spherical and slightly flattened, with a medium weight of 140g/fruit. One plant can produce 5–5.5 kg of fruits. Minaret F1, as Siriana F1, is a fast growing cultivar, with dark red, firm fruits that can weight 180–200 g/fruit. HTP F1 has a short growth cycle; the fruits are pink, firm, fleshy, and juicy with a medium weight of one fruit of 170–250 g. Finally, Inima de Bou cultivar produces big fruits (200–260 g) which have the shape of a heart. All of the cultivars can be cultivated in open-field, greenhouses or tunnels.

### 2.2. Microorganisms

Rizotech plus^®^ (MsBiotech, Larino, Italy) was used for the inoculation of tomato plants. The product was kindly provided by MsBiotech, Italy. The microorganisms of this commercial product consist of an arbuscular mycorrhizal fungus (*Glomus* spp.), PGPR (*Pseudomonas* sp., *Bacillus* spp., *Streptomyces* sp.) and a fungus (*Trichoderma* sp.) in different proportions. The dosage used in our experiments, considering the type of the soil used, was the maximum recommended by the manufacturer, namely 60 kg·ha^−1^. 

### 2.3. Experimental Design

The experiment was carried out at the experimental station of University of Agricultural Sciences and Veterinary Medicine (UASMV) of Iasi, Romania, in a split plots design, in an individual tunnel of 400 m^2^. The study was conducted over 3 years, 2017–2019, starting from the middle of April until the middle of October of each year. During the experiment period, the recorded mean temperature was of 17.2 °C in 2017, 18.4 °C in 2018 and 18.3 °C in 2019. The sunlight period was of 222 h in 2017, 244 h in 2018 and 213 h in 2019. The registered relative humidity was of 67% in 2017, also in 2019, and of 70% in 2018. Prior to tomato plants growing, the plots were cultivated with cucumber. The soil used in the experiment was alluvial cambic chernozem with the following characteristics: 62% sand, 32% clay, 6% silt, 26.6 g·kg^−1^ organic matter, 28 mg·kg^−1^ N, 32 mg·kg^−1^ P, 224 mg·kg^−1^ K, 41 mg·kg^−1^ CaCO_3_, pH 7.2, EC 478 µS·cm^−1^. The soil was not subjected to any sterilization or fumigation processes. The experiment was done in triplicate, the surface of one repetition being of 3.6 m^2^. A number of 12 plants were used for every surface and cultivar. The plants were cultivated in the superficial layer of the soil, so that the roots remained close to the surface. Each cultivar was inoculated with 60 kg·ha^−1^ Rizotech plus^®^/inoculation following a two-treatment scheme: The first treatment was done in the seedling phenophase (14 day after transplanting) and the second treatment at 14 days after planting in tunnel. The inoculation was done close to the roots of tomato plants and was followed by a deep irrigation to enhance the microorganisms’ transfer to the roots. Uninoculated plants were used as control. During the experiment, plants were irrigated 26 times (one time/week for 2.5 h) with two different amounts of water: 200 m^3^·ha^−1^ (5.200 m^3^/vegetation period) and 300 m^3^·ha^−1^ (7.800 m^3^/vegetation period). Growing practices (training, pruning and treatments for pests and diseases) were applied for all the plants, according to the techniques described by Stoleru et al., 2014 [3]. During the experiment, when fruits were fully ripened (BBCH 805–808), a minimum of three fruits from each cluster (3–5) were collected for further analyses.

### 2.4. Growth Measurements

The length of the plants and the number of the fruits were measured at the end of the experiment. The yield (kg·ha^−1^) was calculated by using the following formula: (plants per ha × fruits per plant × average fruit weight)/1000.

### 2.5. Determination of Lycopene Content

The lycopene was obtained from ripe tomatoes by solvent extraction as follows: fresh tomatoes were first homogenized in a blender until a puree was obtained. A sample of 0.6 g of the mixture was added in a vial containing 5 mL of 0.05% (*w*/*v*) butylated hydroxytoluene (BHT) in acetone, 5 mL of 95% ethanol and 10 mL of hexane. The samples were stirred on a magnetic stirring plate for 15 minutes on ice. After stirring, 3 mL of deionized water were added in every vial and the samples were shaken for an additional 5 min on ice. The vials were left at room temperature for 5 min for the phase separation. The absorbance of the upper layer of hexane was measured at 472 nm by using hexane as blank [13]. The total lycopene content was calculated by using the following formula: lycopene (mg·100 g^−1^) = (E/3.45) × (20/*w*); where E = extinction coefficient; *w* = weight (g) [14]. 

### 2.6. DPPH Radical-Scavenging Activity

The antioxidant activity was evaluated by Trolox equivalent antioxidant capacity method (TEAC) assay. Equal volumes of extract or trolox standard and methanolic solution of DPPH (0.1 mM) were mixed and incubated for 30 min at room temperature, in the dark. After incubation, the absorbance was recorded at 517 nm. The results were expressed as mmol equivalents of trolox (an analog of vitamin E) per 100 g of fresh weight (FW) [15,16].

### 2.7. Determination of Total Phenol Content

The total phenol content was assessed by using the Folin-Ciocalteu method as follows: 100 µL of extract was mixed with 3 mL distilled water and 100 µL Folin-Ciocalteu reagent. The resulting solution was left to rest for 3 min. After that, to each flask, 300 µL of 20% sodium carbonate (*w*/*v*) was added. Gallic acid was used as the standard (0–30 µg·mL^−1^). The absorbance of each solution was measured after 2 h at 760 nm, expressing the results as mg gallic acid per 100 g of fresh weight [17]. 

### 2.8. Statistical Analysis

Experimental results are expressed as means ± SD. The data were statistically evaluated by two-ways ANOVA with replication. Tukey’s test was performed in order to estimate the significant difference between the treatment means and among the cultivars. Differences between groups were considered significant when *p* < 0.05.

## 3. Results and Discussion

### 3.1. The Effect of Microorganisms’ Inoculation and Two Water Treatments on the Length of Tomato Plants

The results showed that the length of the plants was influenced by the presence of microorganisms from Rizotech Plus^®^ product only in the case of HTP F1cultivar watered with 300 m^3^, when significant differences were registered between the inoculated and uninoculated plants (Figure 1). For the rest of the cultivars, the presence of microorganisms influenced the length of the plants but not in a significant way. For instance, the length of the plants of Siriana F1 tomato cultivar inoculated with microorganisms is not significant for both irrigation norms (Figure 1). In general, AMF, PGPR or fungi are known for their ability to promote the growth and development of the plants [18,19,20,21,22,23,24]. Previous studies have demonstrated that the presence of AMF and PGPR in the rhizosphere stimulate the growth in the length of various plants as: stevia [20], corn [18], habanero chilli [25], cucumber [26], sesame [27], winter wheat [28], or rice [29] mostly due to a better uptake of nutrients by the plant roots. 

No significant differences were observed between the length of the inoculated plants when they were irrigated with 200 m^3^ or 300 m^3^ of water/hectare (Figure 1). This suggests the fact that Siriana F1, Minaret F1, HTP F1, and Inima de Bou tomato cultivars, in the case of microorganisms’ fertilization, can be watered with only 200 m^3^·ha^−1^ to get almost the same length of plants as in the case of 300 m^3^·ha^−1^. In the agricultural practice, water consumption is a very important issue. According to FAO, agriculture is the major water consumer worldwide, accounting on average for 70% of the total existing freshwater [30]. Use of decreased amounts of water for irrigation represents a sustainable method of water use in agriculture and is also a cost-saving strategy for farmers. Therefore, in the case of the length of the four tomato cultivars, this strategy can be applied.

However, significant differences were noticed between inoculated cultivars in both of the water treatments. Thus, the length of the plants of Siriana F1, HTP F1 and Inima de Bou cultivars was significant bigger than that of the plants of Minaret F1 cultivar, irrespective of the amount of water applied. These differences can be attributed to the fact that Minaret F1cultivar has a semi-determinate growth as compared to the other three cultivars used in the experiments which can grow in an indeterminate manner.

### 3.2. The Effect of Microorganisms’ Inoculation and Two Water Treatments on the Number of Fruits and Yield of Tomato Plants

The presence of the microorganisms was of a significant importance only for the tomato plants belonging to Minaret F1 cultivar for which the number of fruits was significantly bigger as compared with the number of the fruits of uninoculated plants, regardless of the amount of water used in the experiment (Figure 2). For the rest of the cultivars, no significant differences were seen between the number of the fruits of inoculated and uninoculated plants, even though for most of the inoculated plants the registered number of fruits was bigger than that of the uninoculated tomato plants. Positive effects of the AMF-bacteria inoculation on the number of fruits have also been observed by Todeschini et al., 2018 [31] in strawberries or by Bona et al., 2017 [32] in tomatoes.

Regarding the effects of the two water treatments, it was observed that the fruit number of the inoculated tomato plants cultivars was not influenced by the amount of water used in the experiment. No significant differences were registered between the number of the fruits of the inoculated tomato plants watered with 200 or 300 m^3^·ha^−1^ (Figure 2). Therefore, Siriana F1, Minaret F1, HTP F1, and Inima de Bou tomato cultivars in the condition of fertilization with Rizotech plus^®^, can be watered with only 200 m^3^·ha^−1^ to obtain the same number of fruits as in the case of a 300 m^3^ ·ha^−1^ water regime.

As in the case of the length of plants and the number of fruits, the yield was not influenced by the amount of water used for irrigation. Therefore, the yield for the inoculated tomato plants watered with 200 m^3^·ha^−1^ was not significantly different than that of the tomato plants irrigated with 300 m^3^·ha^−1^ of water, regardless of the cultivar used in the experiment (Figure 3). However, significant differences were registered between the inoculated and uninoculated plants on both of the water treatments. Thus, the yield of the plants inoculated with Rizotech plus^®^, irrigated with 200 or 300 m^3^·ha^−1^ of water/application belonging to Siriana F1, Minaret F1 and HTP F1 cultivar was significantly bigger than that of the corresponding uninoculated plants. No differences were noticed between the inoculated and uninoculated Inima de Bou tomato cultivar irrespective of the amount of water used for irrigation (Figure 3).

Moreover, it was observed that between the inoculated cultivars, the yield of the inoculated tomato plants belonging to HTP F1 cultivar was significantly bigger (161.7 t·ha^−1^, respectively 166.25 t·ha^−1^) as compared with that of Inima de Bou cultivar (132.7 t·ha^−1^, respectively 128.11 t·ha^−1^) at both of the water amounts (200 and 300 m^3^·ha^−1^) used in the experiments. No significant differences were noticed between the other inoculated cultivars (Figure 3). 

Therefore, the inoculation of the four tomato cultivars used in the experiments with AMF–bacteria–fungus had positively influenced the shoot length, the number of the fruits and the yield of the plants. While increases in shoot length are a very well-known consequence of microorganisms’ action, their effects on the number of fruits and yield is less studied. In greenhouse experiments, the yield of the tomato plants seems to be enhanced by the microorganisms’ inoculation as demonstrated by Hart et al., 2015 [33], by Salvioli et al., 2012 [34] or by Conversa et al., 2013 [35]. Regarding the number of fruits and the yield obtained in the protected area or field due to biofertilizer application, the results largely vary along with the plant species and cultivars, the microorganisms used for inoculation, the interactions between the rhizospheric organisms, the physical and chemical conditions of the soil, as well as with the environmental factors like temperatures or precipitations [36]. In our study, the microorganisms from Rizotech plus^®^ product inoculated to Siriana F1, Minaret F1 and HTP F1 tomato cultivars grown in tunnel, significantly increased the yield of the plants as compared to the uninoculated tomato plants, irrespective of the water amount used in the experiments, demonstrating the importance of their presence on the plant’s root.

### 3.3. The Effect of the Microorganisms’ Inoculation and the Two Water Treatments on the Lycopene, Polyphenols Content and Antioxidant Activity of Tomatoes

The data showed that the efficiency of microorganism’s treatments is dependent on the amount of water used and on the tomato cultivar when it comes to the tomato fruit’s quality such as lycopene, polyphenols and antioxidant activity (Table 1).

Lycopene represents 60–74% of the carotenoids present in tomatoes, being an essential nutrient for the human diet due to its high antioxidant capacity [37,38]. Studies have reported that due to the protective effects of the carotenoids, tomato consumption decreases the risk of cardiovascular disease, atherosclerosis, and cancer, as well as regulating the immune system [39,40,41]. The lycopene content of tomato fruits can vary with the cultivar, the environmental factors or the presence of microorganisms [42]. 

In the samples measured in this study, the lycopene content of the tomato fruits belonging to Siriana F1 and HTP F1 cultivars inoculated with Rizotech plus^®^, regardless of the water used for irrigation, also for Inima de Bou cultivar watered with 300 m^3^ water/ha, was significantly higher (Siriana F1—11.16, 11.63 mg·100 g^−1^ FW; HTP F1—12.09, 13.02 mg·100 g^−1^ FW; Inima de Bou—13.12 mg·100 g^−1^ FW) than that of the tomato fruits from the corresponding uninoculated plants (9.01, 9.81 mg·100 g^−1^ FW; 11.09, 12 mg·100 g^−1^ FW; and 11.35 mg·100 g^−1^ FW respectively), under the same water regime (Table 1). We can presume that the presence of the microorganisms contributed to the increase of the lycopene content of the tomatoes, probably due to the nutrient status improvement of the plants. Studies have shown that increases in potassium or phosphorus content enhance the lycopene content in tomato fruits [43]. Taking into account that Rizotech plus^®^ contains microorganisms that are known for promoting the growth and development of plants, this hypothesis can be taken into consideration. No significant differences in lycopene were registered between the tomatoes of Minaret F1 (at 200 or 300 m^3^ water/ha) or Inima de Bou (at 200 m^3^ water/ha) inoculated cultivars and the tomatoes of the corresponding uninoculated plants.

Regarding the influence of the amount of water applied on the lycopene content of the tomato fruits, no significant differences were seen between the inoculated plants of Siriana F1 or Minaret F1 cultivars irrigated with 200 or 300 m^3^ water·ha^−1^ (Table 1). These findings are in accordance with the results regarding the length of the plants, the number of fruits, and the yield when no differences were seen between the tomato plants watered with the two water treatments. Anyway, the lycopene content of the tomato fruits of HTP F1 and Inima de Bou inoculated cultivars was significantly higher when water was used in an amount of 300 m^3^·ha^−1^ as compared with 200 m^3^·ha^−1^ (Table 1). According to Grolier et al., 1999 and Atkinson et al., 2011, water stress can reduce the lycopene content in tomato [44,45]. Therefore, in our study, probably, the irrigation of HTP F1 and Inima de Bou tomato cultivar with 200 m^3^·ha^−1^ represented a water stress condition for the lycopene production. However, at 200 m^3^ water·ha^−1^, the best lycopene content was recorded for the tomato fruits of HTP F1 cultivar (13.02 mg·100 g^−1^ FW), while at 300 m^3^·ha^−1^ for Inima de Bou cultivar (13.12 mg·100 g^−1^ FW). The lowest content regardless of the water amount (200 or 300 m^3^·ha^−1^) used was registered for Minaret F1cultivar (11.03 mg·100g^−1^ FW and 11.31 mg·100 g^−1^ FW respectively). Considering the results of the lycopene content, we can conclude that the efficiency of Rizotech plus^®^ product on this parameter varied along with the tomato cultivar used in the experiment and the water amount applied. 

Polyphenols are known for their antioxidant capacity, being very important in the human diet for lowering the risk of cardiovascular disease or cancer or for increasing the resistance of low-density lipoprotein (LDL). As in the case of lycopene, the polyphenol content in plants depends on the cultivation and harvesting conditions such as growing conditions, degree of ripeness and plant variety [46,47]. 

The presence of the microorganisms significantly affected the polyphenols content of the tomatoes from Minaret F1 and Inima de Bou cultivars, at both 200 and 300 m^3^·ha^−1^, when a significant higher value was recorded as compared to the uninoculated plants, but had no effect on Siriana and HTP F1 cultivars, regardless of the amount of water used, when no differences between the inoculated and the uninoculated plants were seen (Table 1). As in the case of the length of the plants, the number of fruits and the yield, no differences in polyphenol content could be seen between the two water treatments used in the experiments for the plants inoculated with Rizotech plus^®^. Therefore, for the cultivars inoculated with microorganisms, by using a lower amount of water for irrigation (200 m^3^·ha^−1^), it can get the same polyphenol content as in the case of a higher amount (300 m^3^·ha^−1^). The highest content of polyphenols at a water regime of 200 m^3^·ha^−1^was registered for Minaret F1 and Inima de Bou cultivars (2805.128 and 2423.26 mg·100 g^−1^ FW), between which no significant differences were observed. At this water amount, significant differences were recorded only between the polyphenol content of the Minaret F1cultivar and Siriana F1 (2283.38 mg·100 g^−1^ FW), as well as HTP F1 (2184.21 mg·100 g^−1^ FW) cultivars (Table 1). At 300 m^3^·ha^−1^, the highest content was registered for the same Minaret F1 and Inima de Bou cultivars, but also for HTP F1 cultivar (2820.385, 2425.239 and 2382 mg·100 g^−1^ FW respectively). Significant differences were seen only between Minaret F1 and Siriana F1 (2322.216 mg·100 g^−1^ FW) cultivars. No differences in the polyphenol content were registered for the rest of the cultivars.

Finally, the antioxidant activity of HTP F1 and Inima de Bou cultivars was significantly influenced by the presence of the microorganisms regardless of the water amount used in experiment. For these cultivars, the antioxidant activity was significantly higher as compared to the uninoculated plants (Table 1). Similarly, Ordookhani and Zare, 2011, reported a higher level of antioxidant activity in the tomato fruits of the plants inoculated with a combination of PGPR and AMF than in the case when no inoculation was considered [48]. 

For Minaret F1 cultivar, the presence of the microorganisms resulted in a higher antioxidant activity only in the case of the plants irrigated with 300 m^3^·ha^−1^ (94.887 mmol Trol·100 g^−1^ FW— inoculated plants vs. 85.93 mmol Trol·100 g^−1^ FW—uninoculated plants), but no differences were recorded between the inoculated and uninoculated plants watered with 200 m^3^·ha^−1^ or between the Siriana F1 cultivar plants irrespective of the water treatment applied. The antioxidant activity was not influenced by the water amount used in the experiments, for three of the cultivars analyzed. Therefore, no differences in the antioxidant activity were seen between the inoculated tomato plants of Siriana F1, HTP F1 and Inima de bou cultivars irrigated with either 200 or 300 m^3^·ha^−1^. Only in the case of the inoculated Minaret F1 cultivar, the antioxidant activity of the plants watered with 300 m^3^·ha^−1^ (94.887 mmol Trol·100 g^−1^ FW) was significantly higher than that of the plants irrigated with 200 m^3^·ha^−1^ (87.563 mmolTrol·100g^−1^FW). The best antioxidant activity in the case of 200 m^3^·ha^−1^ irrigation was registered for the inoculated plants of HTP F1 cultivar (94.05 mmol Trol·100 g^−1^ FW), followed by Minaret F1 (87.563 mmol Trol·100 g^−1^ FW), Inima de Bou (86.96 mmol Trol·100 g^−1^ FW), and Siriana F1 (81.39 mmol Trol·100 g^−1^ FW) cultivar. Significant differences were seen only between the antioxidant activity of inoculated Siriana F1 and HTP F1 cultivar. When the water treatment was 300 m^3^·ha^−1^, the best antioxidant activity was recorded for Minaret F1 (94.887 mmol Trol·100 g^−1^ FW) and HTP F1 (93.97 mmol Trol·100 g^−1^ FW) followed by Siriana F1 (85.34 mmol Trol·100 g^−1^ FW) and Inima de Bou (83.92 mmol Trol·100 g^−1^ FW) inoculated cultivars (Table 1).

## 4. Conclusions

This study shows that the microorganisms from Rizotech plus^®^ product had a positive effect on the growth of tomato plants belonging to Siriana F1, Minaret F1, HTP F1, and Inima de Bou cultivars, influencing the length of the plants, the number of fruits, and the yield. Moreover, the presence of the microorganisms enhanced the tomato fruits quality such as lycopene, polyphenol and the antioxidant activity which are very important in the human diet. The efficiency of Rizotech plus^®^ application is however dependent on the cultivar and the amount of water used. Anyway, the results definitely demonstrate that the Rizotech plus^®^ inoculation significantly increased the yield of Siriana F1, Minaret F1 and HTP F1 cultivars as compared to the uninoculated plants, regardless of the water amount used in the experiment. Moreover, the results strongly suggest that for the irrigation of Siriana F1, Minaret F1, HTP F1, and Inima de Bou cultivars, inoculated with Rizotech plus^®^, a lower amount of water (200 m^3^·ha^−1^) can be used to get the same length of plants, number of fruits and yield as in the case of a higher amount of water (300 m^3^·ha^−1^). In the case of lycopene, polyphenols and antioxidant activity, this situation is cultivar- and water-dependent. The results obtained in this study give new information about the functionality and performance of the microorganisms from Rizotech plus^®^ product when applied to different tomato cultivars, in the condition of two different water regimes, contributing to a better characterization of it and maybe to a more efficient use in agriculture to get optimum results. Further studies should be conducted to see how other important parameters such as the soil type affects the efficiency of this microorganisms-based product.

## Figures and Tables

**Figure 1 microorganisms-07-00706-f001:**
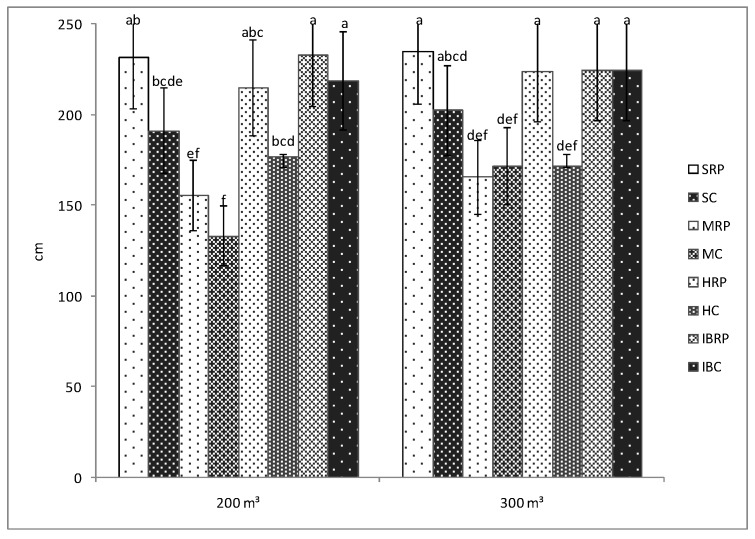
Length of the plants of four tomato cultivars (Siriana F1—S, Minaret F1—M, HTP F1—H, and Inima de Bou—IB) in the presence (RP) and absence (C) of Rizotech plus^®^ and two water treatments (200 m^3^ and 300 m^3^ water/hectare). Different letters mean significant differences between the treatments, according to Tukey post hoc test (*p* < 0.05).

**Figure 2 microorganisms-07-00706-f002:**
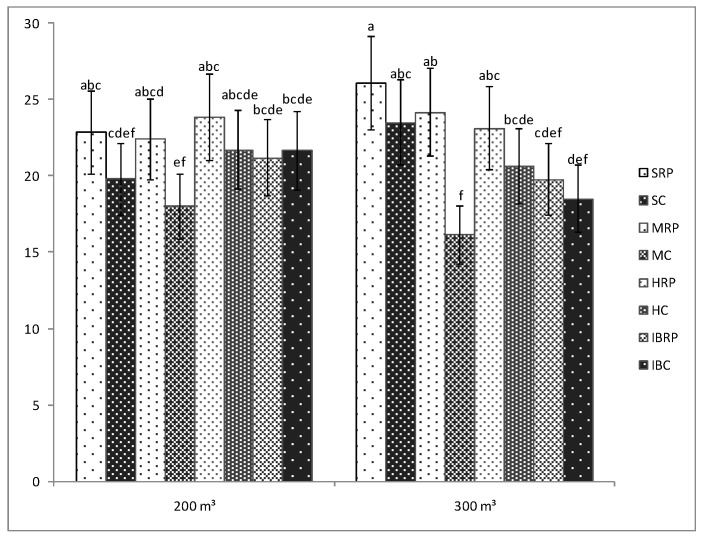
Number of fruits of four tomato cultivars (Siriana F1—S, Minaret F1—M, HTP F1—H, and Inima de Bou—IB) in the presence (RP) and absence (C) of Rizotech plus^®^ and two water treatments (200 m^3^ and 300 m^3^ water/hectare). Different letters mean significant differences between the treatments, according to Tukey post hoc test (*p* < 0.05).

**Figure 3 microorganisms-07-00706-f003:**
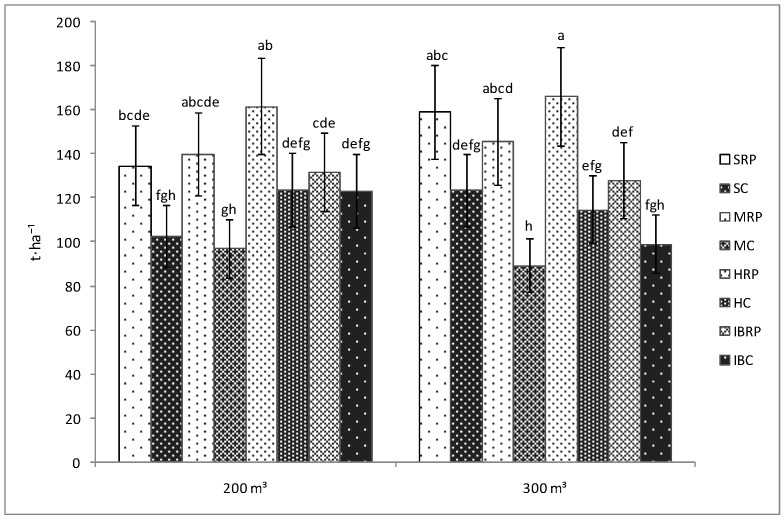
Yield of four tomato cultivars (Siriana F1—S, Minaret F1—M, HTP F1—H, and Inima de Bou—IB) in the presence (RP) and absence (C) of Rizotech plus^®^ and two water treatments (200 m^3^ and 300 m^3^ water/hectare). Different letters mean significant differences between the treatments, according to Tukey post hoc test (*p* < 0.05).

**Table 1 microorganisms-07-00706-t001:** Effects of the microorganisms’ inoculation and the two water treatments on the antioxidant activity, polyphenol and lycopene content of tomatoes.

Treatment	Lycopene (mg·100 g^−1^ FW)472 nm	Polyphenol (mg·100 g^−1^ FW)	Antioxidant Activity (mmol Trol·100 g^−1^ FW)
SRP 200	11.16 ± 0.7 de	2283.38 ± 285.6 bcde	81.39 ± 4.5 cdef
SRP 300	11.63 ± 0.7 bcd	2322.216 ± 410.8 bcd	85.34 ± 4.5 cd
SC 200	9.01 ± 0.1 h	2165.28 ± 280.2 bcde	76.63 ± 5.5 efg
SC 300	9.81 ± 0.2 g	2228.186 ± 362.5 bcde	82.44 ± 1.9 cdef
MRP 200	11.03 ± 0.1 def	2805.128 ± 286.8 a	87.563 ± 5.6 bc
MRP 300	11.31 ± 0.3 de	2820.385 ± 412.9 a	94.887 ± 2.8 a
MC 200	10.39 ± 0.5 fg	1863.402 ± 141.2 de	83.85 ± 3.8 cde
MC 300	10.89 ± 0.1 ef	1882.277 ± 96.4 de	85.93 ± 4.6 cd
HRP 200	12.09 ± 0.5 b	2184.21 ± 157.4 bcde	94.05 ± 4.1 ab
HRP 300	13.02 ± 0.3 a	2382.274 ± 449.2 abc	93.97 ± 4.9 ab
HC 200	11.09 ± 0.3 de	1808.253 ± 88.2 e	70.39 ± 2.5 gh
HC 300	12 ± 0.4 bc	1988.377 ± 156.7 bcde	65.12 ± 4.8 h
IBRP 200	11.5 ± 0.2 bcde	2423.26 ± 306.2 ab	86.96 ± 3.7 bc
IBRP 300	13.12 ± 0.2 a	2425.239 ± 352 ab	83.92 ± 3.5 cde
IBC 200	11.28 ± 0.3 de	1926.223 ± 104.9 cde	78.71 ± 7.1 def
IBC 300	11.35 ± 0.6 cde	1942.296 ± 108.9 cde	75.94 ± 2.9 fg

Different letters mean significant differences between the treatments, according to Tukey posthoc test (*p* < 0.05). S = Siriana F1; M = Minaret F1; H = HTP F1; IB = Inima de Bou; RP= Rizotech plus^®^; C = control; 200 = 200 m^3^ water·ha^−1^; 300 = 300 m^3^ water·ha^−1^.

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
