# Peer review of "The Effects of a Microorganisms-Based Commercial Product on the Morphological, Biochemical and Yield of Tomato Plants under Two Different Water Regimes"

_microorganisms, 2019, doi:10.3390/microorganisms7120706_

Round 1
Reviewer 1 Report
Review: The effects of a microorganisms-based commercial 2 product on the morphological, biochemical and yield 3 of tomato plants under two different water regimes.
The manuscript deals with the main goals of modern horticulture, namely application pro-ecological techniques for economically important crops to affect plant growth and improve the biological quality of yield. Generally, the manuscript presents novelty and merits. Tomato is one of the most important crops worldwide, effective microorganisms are intensively investigated as an alternative for chemical fertilizers and plant protection chemicals. The manuscript is generally well written.
The introduction is based on proper and up-to-date references. Material and methods – The authors stated that the inoculum was placed to the soil twice: 14 days after transplanting, and 14 days after planting in the tunnel. Did the Authors control the establishment/existence/ intensify of AMF inoculation during the experiment? Inoculation’s effectiveness depends on many inoculum-soil-plant factors and sometimes is not effective because of the ‘natural’ microbiota competition. I think, that it is misusing the assumption that inoculation guarantees the mycorrhiza establishment and maintenance (see doi:10.1111/jam.14247). Please, respond to this comment as additional information within the M&M chapter.
There is a lack of information about the method of collecting fruits for analyses: the date of fruit collection (from which cluster?), the stage of ripening, the fruit number per cluster/plant, etc.
Also, there is a lack of information on growing practices, like training/pruning (common in tomato under covers), pollination (bumblebees?), etc. These practices affect plant development in a highly cultivar-dependent manner, and they – if applied – could affect the results of the experiment.
The height of plants, number of fruits per cluster or per plant, the yield and chemical composition (especially lycopene content) are also genetically determined. I think that the cultivar’s description in M&M chapter should be supplemented with these cultivars' characteristics to enable proper interpretation of the results.
Discussion is clear and generally supported by experimental evidence.
In my opinion, the manuscript needs some amendments and it should be checked concerning style or typographical errors (for example ABSTRACT – “...to a better characterization of it and maybe to a better use in agriculture for better results.”
Based on the above opinion I recommend to accept the manuscript after a minor revision.
Author Response
RESPONSE TO REVIEWER 1 (GREEN)
Comment 1: Material and methods – The authors stated that the inoculum was placed to the soil twice: 14 days after transplanting, and 14 days after planting in the tunnel. Did the Authors control the establishment/existence/ intensify of AMF inoculation during the experiment? Inoculation’s effectiveness depends on many inoculum-soil-plant factors and sometimes is not effective because of the ‘natural’ microbiota competition. I think, that it is misusing the assumption that inoculation guarantees the mycorrhiza establishment and maintenance (see doi:10.1111/jam.14247). Please, respond to this comment as additional information within the M&M chapter.
Response 1:We did not control the establishment/existence/ intensify of AMF inoculation during the experiment, because we wanted to have a very close image about what is happening with the tomato plants when a microorganisms-based commercial product is applied in the same manner as farmers are doing it in the field, and when are irrigated with two different water regimes. It is true that many factors can influence the effectiveness of the inoculum, but in our experiments, taking into account that we had controlled conditions, also uninoculated plants as control, irrigated with both of the water regime; we can assume that the differences are because of the presence of the applied microorganisms. To be clear, the following statement was added to Materials and Methods chapter, line 105: “The soil was not subjected to any sterilization or fumigation processes.”
Comment 2:There is a lack of information about the method of collecting fruits for analyses: the date of fruit collection (from which cluster?), the stage of ripening, the fruit number per cluster/plant, etc.
Response 2:Information about the method of collecting fruits for analyses were added within the Materials and Methods chapter (lines 114-116) as follows: “During the experiment, when fruits were fully ripened (BBCH 805 - 808), a number of minimum 3 fruits from each cluster (3-5) were collected for further analyses.”
Comment 3: Also, there is a lack of information on growing practices, like training/pruning (common in tomato under covers), pollination (bumblebees?), etc. These practices affect plant development in a highly cultivar-dependent manner, and they – if applied – could affect the results of the experiment.
Response 3:Information about the growing practices were added within the Materials and Methods chapter, lines 112-114: “Growing practices (training, pruning and treatments for pests and diseases) were applied for all the plants, according to the techniques described by Stoleru et al., 2014 [3].”
Comment 4: The height of plants, number of fruits per cluster or per plant, the yield and chemical composition (especially lycopene content) are also genetically determined. I think that the cultivar’s description in M&M chapter should be supplemented with these cultivars' characteristics to enable proper interpretation of the results.
Response 4:Additional information about the cultivars’ characteristics have been added (lines 86-92):“SirianaF1 is an early growing cultivar, known to produce its first fruits after ~100 days. The fruits are red, spherical and slightly flattened, with a medium weight of 140g/fruit. One plant can produce 5 – 5.5 kg of fruits. Minaret F1, as Siriana F1, is a fast growing cultivar, with dark red, firm fruits that can weight 180 – 200 g/fruit. HTP F1 has a short growth cycle; the fruits are pink, firm, fleshy and juicy with a medium weight of one fruit of 170 – 250 g. Finally, Inima de Bou cultivar produces big fruits (200 – 260g) which have the shape of a heart.”
Comment 5: In my opinion, the manuscript needs some amendments and it should be checked concerning style or typographical errors (for example ABSTRACT – “...to a better characterization of it and maybe to a better use in agriculture for better results.”
Response 5:As it has been suggested, the following edits have been done:
Line 6: Affiliation number from “Vasile Stoleru2” was changed to “Vasile Stoleru1”
Line 16: “proving” was replaced with “providing”
Line 18: “The uses” was replaced with “The use”
Line 19: “product” was replaced with “products”, “are” was replaced with “is”
Line 33: “give” was replaced with “gives”
Line 35: “grown in tunnel” was added
Line 36: “better” from “better use” was replaced with “more efficient”
Line 36: “better” from “better results” were replaced with “to achieve optimum”
Line 61: “protein” was replaced with “proteins”
Line 93: “solarium” was replaced with “tunnels”
Line 106: “harvested” was deleted
Line 122: “follow” was replaced by “follows”
Line 130: “formulas” was replaced with “formula”
Line 139: “follow” was replaced by “follows”
Line 147: “Tukey tests were” was replaced with“Tukey’s test was”
Line 154: “m3” was replaced with “m3”
Line 170: “suggest” was replaced with “suggests”
Line 171: “microorganism” was replaced by “microorganisms’ ”
Line 177: “fourth” was replaced with “four”
Line 191: “for” was replaced with “of”
Line 208: “significant” was replaced with “significantly”
Line 212: “significant” was replaced with “significantly”
Line 237: “tunnels” was replaced with “tunnel”
Line 248: “decrease” was replaced with “decreases”
Line 266: “microorganism’s” was replaced with “microorganisms’ ”
Line 267: “contents” was replaced with “content”
Line 272: “difference” was replaced with “differences”
Line 300: “used in experiment” from “cultivars used in experiment inoculated with microorganisms” was deleted
Line 301: “of the using” from “as in the case of the using of a higher amount” was deleted
Line 308: “between cultivars” was deleted from “between cultivars were seen only between..”
Line 309: “cultivar” was replaced with “cultivars”
Line 316: “than those with no inoculation” was reworded “than in the case when no inoculation was considered”
Line 352: “better” was replaced with “a more efficient”; “for better results” was replaced with “to get optimum results”
Line 360: “their” was replaced with “her”
Line 361: “®” was added to “Rizotech plus”

Reviewer 2 Report
Nice work, but can be improved in several aspects:
A moderate editing for the writing will improve the flow and remove unnecessary wording, example The authors uses the word “anyway” often where it is not needed. Also used “on the other hand” s used twice in the same sentence in page 2. The authors uses some strong language that is not needed, example, in the abstract, the authors say “the organic production of tomato is mandatory”, I do not why it mandatory?The experiment was conducted in small plots, however, the authors report the data as in kg.ha-1. This can be misleading, I would rather see the results based on the number of plants used in the study (12), to give the reader an actual results. In the methods, is the commercial fertilizer application rate is what was recommended by the manufacturer? Figures has no Y-axis legend. The order of the different tomato cultivars in the figure caption differ from the order in the graph itself and in the text, this makes it harder for the reader. If the authors be consistent, that would be best. Also the shading of the histogram is very close and hard to see in the black/white print. In the results, line 149 – 152, I am not sure how the authors made that conclusion about the root system. Did they have any measurements (quantitative or qualitative) of the root system at the end of the experiment? Any supporting data or literature to back up this statement would be needed. Line 279 – 284, the comparison of lycopene content of different cultivars in other studies is irrelevant. Different cultivars/growth conditions will have different effect of the tomato fruits.
Author Response
RESPONSE TO REVIEWER 2 (BLUE)
Comment 1:A moderate editing for the writing will improve the flow and remove unnecessary wording, example The authors uses the word “anyway” often where it is not needed. Also used “on the other hand” s used twice in the same sentence in page 2.
Response 1:As it has been suggested, the following edits have been done:
Line 26: “Anyway” was replaced with “Also,”
Line 29: “Also” was replaced with “Moreover”
Line 130: “w = weight (g); E = extinction coefficient” was reordered “E = extinction coefficient; w = weight (g)”
Line 181: “Anyway” from “Anyway, these differences” was deleted. After deletion the phrase starts with “These differences”
Line 285: “Therefore, in the case of the lycopene content” was replaced with “Considering the results of the lycopene content, we can conclude that”
Line 286: “on this parameter” was added after “the efficiency of Rizotech plus® product”
Line 327: “Anyway” from “Anyway, the best antioxidant activity” was deleted
Comment 2:The authors uses some strong language that is not needed, example, in the abstract, the authors say “the organic production of tomato is mandatory”, I do not why it mandatory?
Response 2:The phrase “In this context the organic production of tomato is mandatory.” was reworded: “In this context the organic production of tomatoes should be taken more into consideration.” (Line 17-18)
Comment 3: The experiment was conducted in small plots, however, the authors report the data as in kg.ha-1. This can be misleading, I would rather see the results based on the number of plants used in the study (12), to give the reader an actual results.
Response 3:Taking into account that we tested a commercial product, we preferred to express the results in kg·ha-1, so that they would be useful for farmers and stakeholders.
Comment 4:In the methods, is the commercial fertilizer application rate is what was recommended by the manufacturer?
Response 4:The manufacturer recommendation is between 30 and 60 kg/ha depending on the plant species and the yield. Taking into account that, as far as we know, there are no studies about the effects of Rizotech plus® on the selected tomato cultivars and soil type, we preferred to use the maximum recommended dosage (60 kg·ha-1). To be clearer, additional information was added to the chapter 2.1 (Line 98-99): “The dosage used in our experiments was the maximum recommended by the manufacturer, namely 60 kg·ha-1.”
Comment 5:Figures has no Y-axis legend.
Response 5:Y-axis legend was added to Figure 1 (Lines 164) and Figure 3 (Lines 216).
Comment 6:The order of the different tomato cultivars in the figure caption differ from the order in the graph itself and in the text, this makes it harder for the reader. If the authors be consistent, that would be best.
Response 6:The order of the tomato cultivars in the figure caption and text were changed as follows:
Line 165: “Figure 1.The length of the plants of four tomato cultivars (Siriana F1- S, HTP F1 - H, Minaret F1– M and Inima de Bou – IB) tomato cultivars in the presence (RP) and absence (C) of Rizotech plus® and two water treatments (200 m3 and 300 m3 water/application). Different letters mean significant differences between the treatments, according to Tukey posthoc test (p<0.05).” was reworded “The length of the plants of four tomato cultivars (Siriana F1- S, Minaret F1– M, HTP F1 - H and Inima de Bou – IB) in the presence (RP) and absence (C) of Rizotech plus® and two water treatments (200 m3 and 300 m3 water/application).”
Line 171: “HTP F1, Minaret” was reordered “Minaret F1, HTP F1”
Line 198: “Inima de Bou and HTP F1” was reordered “HTP F1 and Inima de Bou”
Lines 202-204: “Figure 2.The number of fruits of four tomato cultivars (Siriana F1- S, HTP F1 - H, Minaret F1– M and Inima de Bou – IB) tomato cultivars in the presence (RP) and absence (C) of Rizotech plus® and two water treatments (200 m3 and 300 m3 water/application).” was reworded “Figure 2.The number of fruits of four tomato cultivars (Siriana F1- S, Minaret F1– M, HTP F1 - H and Inima de Bou – IB) in the presence (RP) and absence (C) of Rizotech plus® and two water treatments (200 m3 and 300 m3 water/application).”
Line 212: “Minaret, Siriana F1” was reordered “Siriana F1, Minaret F1…”
Lines 217-219: “Figure 3.The yield of four tomato cultivars (Siriana F1 - S, HTP F1 - H, Minaret F1– M and Inima de Bou – IB) tomato cultivars in the presence (RP) and absence (C) of Rizotech plus® and two water treatments (200 m3 and 300 m3 water/application).” was changed “Figure 3.The yield of four tomato cultivars (Siriana F1 - S, Minaret F1– M, HTP F1 - H and Inima de Bou – IB) in the presence (RP) and absence (C) of Rizotech plus® and two water treatments (200 m3 and 300 m3 water/application).”
Line 237: “Minaret F1, Siriana F1” was reordered “Siriana F1, Minaret F1”
Line 276: “Inima de Bou and HTP F1” was reordered “HTP F1 and Inima de Bou”
Line 279: “Inima de Bou and HTP F1” was reordered “HTP F1 and Inima de Bou”
Line 331: “HTP F1 and Siriana F1” was reordered “Siriana F1 and HTP F1”
Line 337: “Minaret F1, Siriana F1” was reordered “Siriana F1,Minaret F1”
Line 343: “Minaret, Siriana” was changed to “Siriana F1,Minaret F1”
Line 345: “Minaret, Siriana” was changed to “Siriana F1,Minaret F1”
Comment 7:Also the shading of the histogram is very close and hard to see in the black/white print.
Response 7:The shadings of Figure 1 (Line 164), Figure 2 (Line 201) and Figure 3 (Line 216) have been changed.
Comment 8:In the results, line 149 – 152, I am not sure how the authors made that conclusion about the root system. Did they have any measurements (quantitative or qualitative) of the root system at the end of the experiment? Any supporting data or literature to back up this statement would be needed.
Response 8:Because of the lack of measurements for the root system, the statement about this issue was deleted (lines 159 – 164): “The reason why the length of the inoculated plants of HTP F1 cultivar watered with 200 m3 was bigger, but not significant, as compared with the plants watered with 300 m3 might be attributed to the fact that because of the less amount of available water, the roots system of the plants was less developed, decreasing the space for microorganisms colonization, also, their density and by default the availability and amount of substances needed for promoting the growth and development.”
Comment 9:Line 279 – 284, the comparison of lycopene content of different cultivars in other studies is irrelevant. Different cultivars/growth conditions will have different effect of the tomato fruits.
Response 9:As suggested, the comparison of the lycopene content of different cultivars in other studies was deleted (Lines 285-290): “Anyway the lycopene content of the varieties used in this study are far higher than those obtained by Bona et al, 2017 for the tomato plants from TC2000 variety inoculated with either a combination of AM fungi and Pseudomonas sp. Strain 19Fv1T (2.6758 mg·100g-1 FW) or AM fungi and Pseudomonas fluorescens C7 (2.65741 mg·100g-1 FW) , but less than that get by Ordookhani et al., 2010 on Lycopersicon esculentum F1, GS -15 (20.7 mg·100g-1 FW) inoculated with a mix of three different PGPR’s and three different AMF.”
